# All-Trans Retinoic Acid Increases DRP1 Levels and Promotes Mitochondrial Fission

**DOI:** 10.3390/cells10051202

**Published:** 2021-05-14

**Authors:** Bojjibabu Chidipi, Syed Islamuddin Shah, Michelle Reiser, Manasa Kanithi, Amanda Garces, Byeong J. Cha, Ghanim Ullah, Sami F. Noujaim

**Affiliations:** 1Department of Molecular Pharmacology and Physiology, Morsani College of Medicine, University of South Florida, Tampa, FL 33612, USA; mreiser1@usf.edu (M.R.); mkanithi@usf.edu (M.K.); bcha@usf.edu (B.J.C.); snoujaim@usf.edu (S.F.N.); 2Department of Physics, University of South Florida, Tampa, FL 33620, USA; syedislamudd@usf.edu (S.I.S.); gullah@usf.edu (G.U.); 3Lisa Muma Weitz Laboratory for Advanced Microscopy & Cell Imaging, University of South Florida, Tampa, FL 33612, USA; agarces1@usf.edu

**Keywords:** all-trans retinoic acid, DRP1, mitochondrial fusion, mitochondrial fission, mitochondrial network

## Abstract

In the heart, mitochondrial homeostasis is critical for sustaining normal function and optimal responses to metabolic and environmental stressors. Mitochondrial fusion and fission are thought to be necessary for maintaining a robust population of mitochondria, and disruptions in mitochondrial fission and/or fusion can lead to cellular dysfunction. The dynamin-related protein (DRP1) is an important mediator of mitochondrial fission. In this study, we investigated the direct effects of the micronutrient retinoid all-trans retinoic acid (ATRA) on the mitochondrial structure in vivo and in vitro using Western blot, confocal, and transmission electron microscopy, as well as mitochondrial network quantification using stochastic modeling. Our results showed that ATRA increases DRP1 protein levels, increases the localization of DRP1 to mitochondria in isolated mitochondrial preparations. Our results also suggested that ATRA remodels the mitochondrial ultrastructure where the mitochondrial area and perimeter were decreased and the circularity was increased. Microscopically, mitochondrial network remodeling is driven by an increased rate of fission over fusion events in ATRA, as suggested by our numerical modeling. In conclusion, ATRA results in a pharmacologically mediated increase in the DRP1 protein. It also results in the modulation of cardiac mitochondria by promoting fission events, altering the mitochondrial network, and modifying the ultrastructure of mitochondria in the heart.

## 1. Introduction

The mitochondrion is a dynamic organelle within the cell that produces energy as adenosine triphosphate (ATP) [1]. Sustaining functional mitochondria in optimal numbers and morphology is essential to cell survival and plays an important role in modulating a diversity of physiological processes. Mitochondrial self-regulatory mechanisms that include a dynamic interplay of the organelle’s fusion, fission, mitophagy, and biogenesis are but a few factors shown to modulate mitochondrial quality control [2]. Mitochondrial fusion and fission are thought to be necessary for maintaining a robust population of mitochondria. Fusion is a dynamic event that leads to the merging of healthy individual mitochondria in order to form a new elongated organelle. Such an event ensures optimal cellular bioenergetics by producing adequate quantities of ATP to sustain cellular function [3]. Failure to carry out mitochondrial fusion begets impaired morphology and consequentially reduces the organelle’s efficiency. On the other hand, mitochondrial fission supports a healthy mitochondrial structure by eliminating the damaged parts of a mitochondrion [4,5]. The balance of mitochondrial fusion and fission is dynamically regulated. Mitochondrial fusion is controlled by GTPases such as mitofusins (MFN1 and MFN2) on the outer mitochondrial membrane and optic atrophy 1 (OPA1) on the inner mitochondrial membrane. The fission events are controlled by dynamin-related protein (DRP1) and fission protein 1 (FIS1) [4,6,7,8]. The functional quality and architecture of the mitochondrial network is maintained through an appropriate equilibrium of fission and fusion events [9,10]. Damaged mitochondria are selectively removed by the process of mitophagy, which is regulated by the PTEN-induced putative kinase 1 (PINK1)/Parkin pathway [11]. The E3 ubiquitin ligase, Parkin is available in the cytosol in normal conditions, but it is promptly translocated to the mitochondria upon the loss of mitochondrial membrane potential (Δψm) [12,13]. Alterations or mutations in fission and fusion events lead to mitochondrial dysfunction, irregular morphology, and cellular instability. Mitochondrial dysfunction has been implicated in several diseases including diabetes [14,15,16], retinal failure [17,18], liver failure [19], and neurological disorders [20,21,22,23,24,25].

Micronutrient retinoids such as all-trans retinoic acid (ATRA) can promote proper cellular differentiation, which makes them of interest for clinical applications [26,27,28,29,30]. ATRA is obtained in the form of carotenoids from plants or retinyl esters derived from diet and is stored in the liver, lungs, kidneys, and bone marrow [31]. ATRA is a bioactive derivative of Vitamin A and is involved in a broad spectrum of biological processes [32]. ATRA is thought to be important for neurogenesis, carcinogenesis, craniofacial morphogenesis, the formation of body axis and limbs, and the development of the lung, kidney, and eyes [33,34,35,36,37]. ATRA has been therapeutically used to treat acute promyelocytic leukemia [38], acne [39], and aging skin [40]. Additionally, studies are beginning to show that ATRA can modulate mitochondria. For instance, ATRA-treated adipocytes showed increased mitochondrial biogenesis by upregulating genes responsible for mitochondrial DNA replication and transcription [41]**.** Vantaggiato C et al. found that retinoic acid increases neuronal differentiation from induced pluripotent stem cells by elevating the mitochondrial fission protein DRP1 [42]. However, there is no direct evidence of ATRA’s effect on mitochondrial fusion and fission events.

In the heart, mitochondrial homeostasis is critical for sustaining normal function and optimal responses to metabolic or environmental stressors [43]. Disruptions in mitochondrial fission and/or fusion can lead to cellular dysfunction and apoptosis, important causes of cardiac myocyte death, particularly in heart failure [44,45,46,47,48,49,50]. In this study, we investigated the direct effects of ATRA on the mitochondrial network in cell culture and in murine hearts using Western blot, confocal, and transmission electron microscopy, as well as mitochondrial network quantification using agent-based stochastic modeling.

## 2. Methods

### 2.1. Cell Culture

Human embryonic kidney 293 (HEK293) and murine mouse atrial cell lines (HL-1) cells [51] were used in this study. HEK293 cells were cultured and maintained in Dulbecco’s modified Eagle’s cell culture medium (#119955, Gibco, Gaithersburg, MD, USA) supplemented with 5% FBS (#26140095, Gibco, Gaithersburg, MD, USA). HL-1 cells were cultured in Claycomb medium (#51800C, Sigma Aldrich, St. Louis, MO, USA) supplemented with 10% FBS (#TMS-016-B), 0.2 mM norepinephrine (#A0937, Sigma Aldrich, St. Louis, MO, USA), and 5 mM glutamine (#G7513, Sigma Aldrich, St. Louis, MO, USA) [52]. The dose-dependent effect of ATRA on mitochondrial fission and fusion were measured in HEK293 cells incubated with 0.1, 0.5, 1, and 10 µM ATRA for 24 h. Time-dependent effects were measured at 6, 12, and 24 h in HEK293 cells treated with ATRA. HL-1 cardiomyocytes were incubated with 1 µM ATRA for 24 h to analyze the mitochondrial network, fission, and fusion events.

### 2.2. Animals

Animal experiments were approved by the IACUC at the University of South Florida. A total of six, 2-months-old, C57BL/6J mice of both sexes were used. Mice were housed in ventilated racks with ad libitum access to food and water. Three mice were treated with a 200 µL of ATRA (#R2625, Sigma Aldrich, St. Louis MO, USA), (150 mg/Kg, dissolved in corn oil) via i.p. injection. Three animals received i.p. injections of corn oil and served as controls. After 24 h, the hearts were isolated via thoracotomy for the quantification of mitochondrial fission and fusion proteins using Western blot, morphology analysis using TEM, and the colocalization of mitochondrial fusion and fission proteins was conducted with confocal immunofluorescence.

### 2.3. Western Blot

To collect lysates from cell culture after treatment with ATRA, cells were rinsed with phosphate-buffered saline (PBS) and incubated on ice for 10 min in a RIPA lysis buffer (#89900, Thermo Scientifics, Carlsbad, CA, USA) containing a protease inhibitor (#P8340, Sigma Aldrich, St. Louis, MO, USA). Cells were collected into a 1.5 mL Eppendorf tube with a cell scraper and then triturated several times by passing through a 1 mL syringe. The total lysate was centrifuged at 12,000 rpm for 20 min, and the supernatant was collected.

For collecting lysates from the ATRA-treated and control mice hearts, mice tissues were frozen to cryogenic temperatures in liquid nitrogen, and tissues were cryo-powdered with a Bessman tissue pulverizer (Spectrum Med, San Diego, CA, USA). Protein isolation was completed by adding 10 µL/mg of an RIPA lysis buffer to the cryopowder. The total lysate was centrifuged at 12,000 rpm for 20 min, and the supernatant was collected.

Protein concentrations were determined with a BCA assay (Pierce™ BCA Protein Assay Kit; #23225, ThermoFisher, Carlsbad, CA, USA). Total protein (10 µg) was loaded and separated by 4–20% premade SDS-PAGE gels (#4568095, BioRad, Mini-PROTEAN TGX Stain-Free Precast Gels, Irvine, CA, USA), then transferred to a nitrocellulose membrane, and probed with an antibody against MFN1/2 (1 µg/mL; #ab57602; abcam, Cambridge, MA, USA), FIS1 (1:1000; #PA5-22142, Invitrogen, Carlsbad, CA, USA), OPA1 (2 µg/mL; #NB110-55290, NOVUS Biotechnologies, Centennial, CO, USA), DRP1 (1:1000; # #8054, Cell Signaling, Danvers, MA, USA), and GAPDH (1:2000; #sc-137179, Santa Cruz Biotechnology, Santa Cruz, CA, USA) at 4 °C overnight. After washing, the membrane was incubated with secondary fluorescent IRDye 800 CW and 680 CW goat anti-rabbit and anti-mouse antibodies (1:25,000; LiCOR, Nebraska, NE, USA) for 1 h. The final images were detected with LiCOR Odyssey and protein bands were quantified using Imagestudio-LiCOR, software-2019.

### 2.4. Isolation and Purification of Mitochondria from the Mouse Heart

The isolation and purification of mitochondria were carried out at 4 °C for the entire 75 min procedure. Upon completing the procedure, the purified, isolated mitochondria were used within 2 h. Isolation buffer A contained (in mM) 70 sucrose, 210 mannitol, 1 EDTA-Na_2_, and 50 Tris-HCl at pH 7.4, and buffer B contained (in mM) 250 sucrose, 10 HEPES-Na, and 1 EDTA-Na_2_ at pH 7.4. The lower two-thirds of the ventricles were finely minced in 1 mL of buffer A, homogenized (10 rapid strokes) using a Dounce hand homogenizer (#KT885300-0002, Kimble™ Kontes™, Millville, NJ, USA), and transferred into a 1.5 mL Eppendorf tube. For 3 min, the homogenate was centrifuged at 1300× *g*. The supernatant was collected and centrifuged for 10 min at 10,000× *g*. The crude mitochondria pellet was collected and suspended in a 55 μL isolation buffer A. This crude mitochondrial preparation was then overlaid on 3 mL of 30% (*v/v*) Percoll (#GE17-0891-01, GE Healthcare, Danderyd, Sweden) in buffer B and centrifuged for 45 min at 50,000× *g*. The resulting three distinct layers, M1, M2, and M3 [53,54], were isolated, transferred to 1 mL of isolation buffer A, and further centrifuged for 5 min at 12,000× *g*. The mitochondrial pellet was carefully washed with 1 mL of isolation buffer A and then resuspended in 500 µL of buffer A for confocal microscopy.

### 2.5. Confocal Microscopy

HL-1 cardiomyocytes were cultured in an eight-well confocal chamber (#155409, LAB-TEK, København, Copenhagen, Denmark) and treated with 1 µM ATRA or vehicle for 24 h. Cells were incubated with 50 nM MitoTracker™ Red (#M22425, Thermo Fisher Scientific, Carlsbad, CA, USA) for 30 min at 37 °C and washed with freshly prepared FluoroBrite Dulbecco’s modified Eagle’s cell culture medium (#119955, Gibco, Gaithersburg, MD, USA) supplemented with 10% FBS (#26140095, Gibco, MD, USA). A laser-scanning confocal microscope (Leica-SP8, Wetzlar, Germany) with a 63× oil objective was used to collect Z-stack images of cells. The mitochondrial network was quantified using MATLAB-based tools described below.

Immunofluorescence in isolated mitochondria was performed as described [55] In short, isolated mitochondria from both control and ATRA-treated mice were incubated with 500 µL of buffer A containing 100 nM MitoTracker™ Red for 1 h at 4 °C. Mitochondria were then washed with PBS, followed by centrifugation at 12,000× *g* for 5 min and being plated dropwise onto coverslips precoated with 0.1% poly-l-lysine (#P8920, Sigma-Aldrich, St. Louis, MO, USA) for 2 h at room temperature, and finally being washed with PBS. The attached mitochondria were fixed in 4% paraformaldehyde for 10 min at room temperature. Coverslips were gently washed three times for 5 min with PBS and permeabilized with 0.1% Triton X-100 in PBS for 10 min prior to being incubated with the primary antibody in PBS with 0.02% Tween (PBS-T) for 90 min at room temperature. The used antibodies were: rabbit monoclonal DRP1 antibody (1:1000; #5391; Cell Signaling, Danvers, MA, USA) and rabbit monoclonal OPA1 antibody (2 µg/mL; #NB110-55290, NOVUS Biotechnologies, Centennial, CO, USA). The primary antibody was removed by washing 3 times for 5 min with PBS-T. The preparations were then incubated with donkey anti-rabbit Alexa-488 conjugated antibody (1:10,000; #A-21206, Life Technologies, CA, USA) on ice for 1 h in PBS-T. The coverslips were then mounted using VECTASHIELD^®^ antifade mounting media (#H1000, Maravai Life Sciences, San Diego, CA, USA) and imaged on a confocal microscope (Leica-SP8, Wetzlar, Germany) using a 63× oil objective. Immunolocalization was quantified between MitoTracker™ Red and DRP1 or OPA1 by analyzing Mender’s (M1 and M2) coefficients using the Coloc 2 plugin Fiji version (2.1.0/1.53c) of ImageJ (NIH, Bethesda, MD, USA) software (https://fiji.sc/, accessed on 12 November 2020), as has been done before [56,57]. M1 and M2 are the summed intensities of colocalized green and red pixels, normalized by total green and red pixels; M1 measures the DRP1 or OPA1 green signal overlapped with MitoTracker™ Red signal and, M2 measures the MitoTracker™ Red signal overlapped with DRP1 or OPA1 green signal [58].

### 2.6. Image Analysis

In living cells, mitochondria exist as highly flexible and dynamic network architectures. Their ability to rapidly change from fully connected to fragmented structures driven by diverse cytosolic conditions and cell states is regulated by microscopic mitochondrial fission and fusion. Cells continuously change the rates of these processes in response to changing energy and metabolic demands to facilitate the redistribution of mitochondria (altering its topology) throughout the cell. Any dysregulation in fission and fusion rates result in the fragmentation of mitochondrial network and are linked to many diseases [59].

In graph theoretic terms, a mitochondrial network is described by nodes (mitochondrion) and edges (connections between individual mitochondria). Each node in the network can have zero (isolated mitochondrion, i.e., no connection with another mitochondrion), one (two connected mitochondria, like o–o where o represents a mitochondrion), two (three mitochondria connected in a linear chain like o–o–o), or three (four mitochondria connected in branched fashion, i.e., o–o–o with the mitochondrion in the middle connected to an additional mitochondrion) neighbors (mitochondria) termed as the degree (k) of the node (Figure 6A). Our analysis showed that nodes with degree four (five mitochondria connected in branched fashion, i.e., o–o–o with the mitochondrion in the middle connected to two additional mitochondria) are rare and are, therefore, ignored. Depending on the pathophysiological state of the cell, a mitochondrial network evolves into different topologies ranging from a fully fragmented state (a network comprising nodes of degree zero or one only) to a network with clusters of varying sizes that are made of loops and branches to a fully connected network of a single cluster. We refer to the largest cluster in the network (the network with the highest number of edges or connections) as a giant cluster and represent the number of edges in the giant cluster by Ng. Thus, in the case of a fully connected network, the size of the giant cluster is the same as the whole network—that is Ng = N, where N is the total number of edges (connections) in the entire mitochondrial network in the cell. Under normal physiological conditions, a mitochondrial network comprises many (often interconnected) clusters with different sizes. Topologically, a mitochondrial network can be uniquely distinguished by various microscopic network parameters such as mean degree <k> (sum of degrees of all nodes in the network divided by N), N_g_, and N_g_/N, as well as features like the distributions of branch lengths (number of edges in a linear branch), loop sizes (number of edges in a cyclic loop, i.e., several mitochondria making a loop), and cluster sizes (number of edges in a cluster, which consist of both linear branches and cyclic loops but is disconnected from the rest of the network).

We used our previously curated pipeline of MATLAB-based tools to extract network parameters from experimental images of mitochondrial networks in control and ATRA-treated HL-1 cells using the following steps [60].

Step 1: We used the MATLAB function *im2bw* to generate a binary image (not shown) from the preprocessed gray scale image (not shown) of the original micrograph (left panels of Figure 5A,B) by applying an appropriate threshold intensity using MATLAB function *graythresh*.

Step 2: A single, one-pixel-thick skeleton image representing mitochondrial network (right panels of Figure 5A,B) was generated from the binary image (using *bwmorph* function from MATLAB) of step 1. Each pixel (representing a single mitochondrion in the right panels of Figure 5A,B) is colored based on whether it has one (red), two (green), or three (blue) neighbors (edges or degree).

Step 3: A skeletonized image from step 2 was converted to an undirected graph to extract network parameters <k>, N_g_, N_g_/N, branch lengths, loop sizes, and cluster sizes (using the MATLAB function *bwlabel*).

### 2.7. Modeling Mitochondrial Network

While the confocal microscopy and TEM are useful in quantifying mitochondrial network parameters, estimating the microscopic rates of fission and fusion are beyond the scope of such analysis. Data-guided stochastic modeling can go a step further and provide a quantitative assessment of how the microscopic fission and fusion rates change in ATRA-treated cells in order to explain the experimental results.

To simulate the mitochondrial network, we used the model described by Sukhorukov et al. [61]. A simplistic mitochondrial network consists of two mitochondria (called a dimer) connected by an edge, referred to as X_1_. In our simulations, a mitochondrial network grows/disintegrates through the two fusion/fission reactions (Figure 6A) given in Equations (1) and (2), respectively (see below). In the first (tip-to-tip) reaction (Figure 6A), the tip (one of the red nodes in Figure 6A) of a dimer, representing a single mitochondrion, fuses with the tip of another dimer to form a linear network of three mitochondria (X_2_) with an internal node (green node in Figure 6A) of degree 2. Note that the fusion of an X_1_ species with the end-node of an X_2_ species or the fusion of the end-nodes of two X_2_ species results in a linear chain of more than three mitochondria with all internal nodes of degree 2 (that is, each internal mitochondrion is connected to one mitochondrion on the right side and one mitochondrion on the left side). In the second (tip-to-side) reaction (Figure 6A), the tip of a dimer (X_1_) fuses with the internal node (green node with degree 2) of X_2_ to form a branched mitochondrial network X_3_ with the internal node (blue node in Figure 6A) of degree 3 (that is, the mitochondrion in the middle is connected to three neighboring mitochondria). Note that the tip-to-side reaction can also occur due to the fusion of the end-node (mitochondrion at the end) of a linear chain (three or more mitochondrion connected in a linear chain) with the internal node (green) of X_2_ species or another linear chain. Similarly, the end-node of a linear chain can react with the other end (tip-to-tip reaction) or the internal nodes (tip-to-side reaction) in the same chain to form cyclic loops. Generally, tip-to-side fusion reactions lead to clusters with loops and branches, whereas tip-to-tip fusion reactions tend to result in more linear chains. While fusion reactions are responsible for making bigger clusters, fission reactions (the reverse arrows in Equations (1) and (2)) disintegrate a given network. In the tip-to-tip fission reaction (Figure 6A and Equation (1) reverse arrow), X_2_ disintegrates into two X_1_ species. In the tip-to-side fission reaction (Figure 6A and Equation (2) reverse arrow), an X_3_ node disintegrates to form one X_1_ and one X_2_ species. All the possible fusion and fission reactions described above can be represented by the following two reaction equations and are shown graphically in Figure 6A.
(1)$$X_{1}+X_{1}\;\begin{array}{c} \overset{a_{1}}{\to }\\ \underset{b_{1}}{\leftarrow } \end{array}\;X_{2}\;\;$$
(2)$$X_{1}+X_{2}\;\begin{array}{c} \overset{a_{2}}{\to }\\ \underset{b_{2}}{\leftarrow } \end{array}\;X_{3}$$
where a_1_ and a_2_ are the rates for the tip-to-tip and tip-to-side fusion reactions, respectively, and b_1_ and b_2_ are the rates for the respective fission reactions. As mentioned above, nodes of degree 4 are extremely rare and therefore ignored in the model [59]. Network edges connecting nodes define the minimal (indivisible) constituents of the organelle. Therefore, all parameters from our simulations were calculated in terms of number of edges in the network.

We implemented our mitochondrial network model as an agent-based model using the Gillespie algorithm [61]. Assuming that we wanted to model a mitochondrial network of size N (N is generally determined by the size of the mitochondrial network retrieved by experimental micrographs, as described above in the Image Analysis section), each of our simulations started with N number of X_1_ species (Table 1). The network was allowed to stochastically evolve through a sequence of fusion and fission processes according to their propensities (probability of a reaction to occur) at a given time step. We ran the algorithm for 10 × N time steps to reach steady state and extracted various network features (<k>, N_g_, branch lengths, etc.) at the end of the simulations using different graph and network algorithms in MATLAB’s graph toolbox. Depending on the used fusion (a_1_ and a_2_) and fission (b_1_ and b_2_) rates (Figure 6A and Equations (1) and (2)), networks of varying properties ranging from one comprising interconnected linear chains and branched clusters to a single fully-connected giant cluster could be generated [61]. We searched for a network with experimentally observed properties (<k>, N_g_, branch length, etc.) [59,61] by varying the ratio of fusion and fission processes using parameters C_1_ = a_1_/b_1_ and C_2_ = a_2_/b_2_ by, respectively, fixing b_1_ = 0.01 and b_2_ = (3/2)*b_1_ and allowing a_1_ and a_2_ to vary. Assuming fixed b_1_ and b_2_ values, each value of a_1_ and a_2_ (and hence C_1_ and C_2_) generates a single mitochondrial network of a fixed topology (for a fixed size N of the initial network). This way, for every (C_1_ and C_2_) pair in the (C_1_ and C_2_) phase space, a mitochondrial network with different topologies can be generated. For every set of C_1_ and C_2_ values, we repeated the simulation 100 times, each time using a different random sequence for the stochastic fusion and fission processes. The network parameters/features averaged over all 100 runs are reported as the final results below (Figure 6). Larger values of C_1_ and C_2_ indicate more frequent tip-to-tip and tip-to-side fusion reactions, respectively, and vice versa. A very small value of C_2_ (or C_1_) results in a network mainly consisting of linear (X_2_ and X_1_) networks (typical of a fragmented network) with small <k> and N_g_/N. A medium value of C_2_ leads to a network having clusters with both branches and loops, whereas a large C_2_ value results in a network having one giant cluster with large <k> and N_g_/N values where nearly all mitochondria are connected in a continuous network.

### 2.8. Transmission Electron Microscopy (TEM)

C57Bl/6J mice were treated with ATRA (150 mg/kg in corn oil, i.p. injection). Control mice were treated with the same volume of corn oil. Twenty-four hours later, 4 mm^3^ of left ventricular walls were collected and trimmed into longitudinal blocks. After overnight storage in 2.5% glutaraldehyde (#16120, Electron Microscopy Sciences, Hatfield, PA, USA), the sections were postfixed in 1% osmium tetroxide (#19150, Electron Microscopy Sciences, PA, USA) for 1 h. The tissue samples were dehydrated through a series of graded ethanol and embedded in Embed-812-resin (#14120, Electron Microscopy Sciences, PA, USA). Semithin sections were cut at 1 µm, mounted on glass slides, and stained with 1% Toluidine Blue O (#AAJ6601514, Fisher Scientific, Tracy, CA, USA). Ultrathin sections were cut at a thickness of 90 nm on an ultramicrotome (Ultracut E Richert-Jung, Vienna, Austria) and stained with 2% uranyl acetate (#22400, Electron Microscopy Sciences, PA, USA) and lead citrate, (#17800, Electron Microscopy Sciences, PA, USA) [62]. Selected ultrathin sections were imaged with TEM (JEOL; JEM-1400, Akishima, Tokyo, Japan) at 10,000X, 25,000X, and 50,000X magnification. Mitochondrial size (average area), number, and circularity were analyzed using Fiji version (2.1.0/1.53c) of ImageJ (NIH, Bethesda, MD, USA) software (https://fiji.sc/, accessed on 12 November 2020). Mitochondrial number, area (μm^2^), and perimeter (μm) were analyzed at 10,000X magnification using the ImageJ software (NIH, USA), as previously done by Demeter-Haludka V et al. [63]. Mitochondrial circularity was also computed as 4π x area/perimeter^2^ [64].

### 2.9. Statistics

Data are presented as mean ± standard error of the mean. Unpaired 2-sample *t*-test or one sample *t*- and Wilcoxon-test were used for statistical analysis.

## 3. Results

### 3.1. All-Trans Retinoic Acid (ATRA) Upregulates the Fission Protein DRP1

We tested the effects of ATRA on DRP1 and the fusion protein OPA1 levels in HEK293 cells using Western blot. ATRA was dissolved in DMSO, and cells were incubated with 0.1, 0.5, 1, and 10 µM ATRA or DMSO vehicle control for 24 h. Figure 1A shows that ATRA resulted in a dose-dependent increase in the levels of DRP1 normalized to GAPDH (* *p* < 0.05 vs. control; *n* = 4 each). However, the levels of OPA1 were not significantly changed (Figure 1B). We also tested the time-dependent effect of ATRA in HEK293 cells. Cells were cultured with 0.5 µM ATRA for 6, 12, and 24 h. DRP1 levels were significantly increased after 24 h of treatment (data not shown). It was also found that 1 µM of ATRA had no effect on mitochondrial fusion proteins MFN1/2 and mitochondrial fission protein FIS1 (Appendix A).

We then tested whether ATRA could lead to DRP1 upregulation in vivo. Two-months-old C57BL/6J mice of both genders were used for ATRA (*n* = 3) and control (*n* = 3) treatments. ATRA (150 mg/Kg) was dissolved in in 200 µL of corn oil and injected i.p. Control mice were treated with the same volume of corn oil [65,66]. After 24 h, total lysates were obtained from both ATRA-treated and control mice hearts. A total of 10 µg of proteins were used to assess DRP1 and OPA1 levels using Western blot. Similarly to the in vitro results, Figure 2 shows that ATRA significantly increased DRP1 protein levels (panel A top and panel B) but not OPA1 levels (panel A bottom and panel C) in mouse hearts (* *p* < 0.05 vs. vehicle). ATRA did not have an effect on MFN1/2 and FIS1 (Appendix A).

### 3.2. Localization of DRP1 and OPA1 Levels in Purified Mitochondria from ATRA-Treated Mouse Hearts

To further validate ATRA’s effect on DRP1 and OPA1, we isolated live mitochondria from ATRA-treated and control mice ventricles using the sucrose and Percoll gradient method [53,54]. The purified and isolated mitochondrial fraction was co-labeled with MitoTracker™ Red and antibodies against DRP1 or OPA1 (Figure 3). The colocalizations of DRP1 and OPA1 were analyzed by calculating Mender’s coefficient using ImageJ (NIH, USA), as previously done by Strack S et al. [67] in a similar experiment that showed DRP1-dependent mitochondrial fragmentation, and our findings were consistent with those results [67]. ATRA caused a significant increase in DRP1 immunofluorescence colocalizing with MitoTracker™ Red signals compared to the control (Figure 3A,B). OPA1 showed no difference between the control and ATRA-treated groups (Figure 3C,D; * *p* < 0.01 vs. control). In order to verify the integrity of our mitochondrial preparations, TEM was performed on a mitochondrial suspension.

### 3.3. Mitochondrial Ultrastructural Changes in ATRA-Treated Mice

We quantified mitochondrial morphology by TEM in the ATRA-treated mouse hearts versus controls. Figure 4A shows micrographs from control and ATRA-treated ventricular tissue sections at 10,000×, 25,000×, and 50,000× magnification. We quantified mitochondrial area, perimeter, circularity, and number in hearts from three ATRA- and three vehicle-treated mice. ATRA significantly decreased the average mitochondrial area (Panel B; *** *p* < 0.001 vs. control) and perimeter (Panel C; *** *p* < 0.001 vs. control), and it significantly increased circularity (Panel D; * *p* < 0.05 vs. control). The number of mitochondria tended to be increased by ATRA but did not reach significance (Panel E; * *p* < 0.05 vs. control).

### 3.4. Modeling Mitochondrial Networks in ATRA-Treated HL-1 Cells

Graph theoretic network parameters have been widely used in computational modeling of mitochondrial fission and fusion [59,68,69,70,71,72]. Here, we applied our image processing toolkit and stochastic network simulation paradigm, as explained above, to process experimental micrographs and model mitochondrial networks in control and ATRA-treated HL-1 cells. HL-1 cell lines are mice atrial myocytes [52]. We chose HL-1 cells because the mitochondria are more abundant, representing approximately one-third of the mass of the heart (35% of the volume of adult cardiomyocytes), and play a critical role in maintaining cellular function. HL-1 cells were treated with vehicle or ATRA (1 µM) for 24 h. After 24 h of treatment, cells were stained with MitoTracker™ Red FM (50 nM; #M22425, Thermo Fisher Scientific, CA, USA) for 30 min, and Z-stack images were collected from live cells using a confocal microscope with a 63x-oil objective.

We analyzed Z-stack images from nine control cells and six ATRA-treated HL-1 cells. After preprocessing each experimental image, we generated a graph of the experimental mitochondrial image (Figure 5A,B) and retrieved microscopic network parameters such as <k>, Ng, Ng/N, loop sizes, branch lengths, and cluster sizes for each case (Figure 5C–I). We found that loop sizes, branch lengths, and cluster sizes were all shifted to smaller values in the ATRA-treated cells (blue) than in the control cells (red), suggesting a relative increase in fission in ATRA-treated cells (Figure 5C–E). Similarly, microscopic network parameters including mean degree, average cluster size, largest cluster size (Ng), and largest cluster size normalized with respect to the total network size (Ng/N) were reduced in ATRA-treated cells (gray bars) compared to control cells (black bars) (Figure 5G–I). However, the mean degree (<k>) and average cluster size were significantly decreased in ATRA-treated HL-1 cells compared to the control. The largest clusters size (Ng) (Figure 5H; *p* = 0.19 vs. control) and the largest cluster size normalized with the total network (Ng/N) tended to decrease but did not reach statistical significance.

After identifying the various network parameters from experimental images, we performed stochastic simulations (Figure 6A) to identify microscopic fusion and fission rates responsible for the network topology in the control and ATRA-treated cells. For a mitochondrial network of size N (approximated from experiential micrograph), we started with N number of X_1_ species and ran the simulations with a given combination of C_1_ and C_2_ values. Over time, the network evolved such that it had different numbers of X_1_, X_2_, and X_3_ species at given iteration numbers. A typical evolution of the network in terms of the distribution of X_1_, X_2_, and X_3_ species as the iteration number increases is shown in Figure 6B for two different combinations of C_1_ and C_2_ values. Simulations were run until a steady state was reached, where the number of X_1_, X_2_, and X_3_ species were not changing anymore (Figure 6B), and parameters were retrieved from the resulting network as reported below.

The above approach allowed us to scan the (C_1_ and C_2_) phase space and find the C_1_ and C_2_ values that resulted in a mitochondrial network with the topology and microscopic parameters, as observed in the control and ATRA-treated cells. As an example, results from simulations (for a network of size *n* = 3000) at fixed C_1_ = 0.0007 (a_1_ = 0.000007, b_1_ = 0.01, and b_2_ = (3/2)*b_1_) but varying C_2_ (tip-to-side fusion reaction rate a_2_) are shown in Figure 6C,D. Increasing the tip-to-side fusion reaction resulted in networks with larger branched clusters with higher mean degree <k>, N_g_/N, cluster sizes, etc. Figure 6C,D clearly show that <k> and N_g_/N both increased as we increased C_2_ (and hence a_2_). These simulations were repeated until we found a combination of C_1_ and C_2_ values that resulted in a network with <k> and N_g_/N values that matched with those experimentally observed within certain tolerance [60].

In Figure 6E, we show the N_g_/N vs. <k> values of mitochondrial network in experimental images from one control and one ATRA-treated cell (red bullets), as well as the fits from the model (black crosses). Both observed and simulated networks showed a significant shift in N_g_/N and <k> towards smaller values in ATRA-treated cells compared to the control cells. We remark that the fit to the data could be further improved by decreasing the increment by which we changed C_1_ and C_2_ values while scanning the parameter space. However, decreasing this increment exponentially increased the computational time. Thus, a compromise between the accuracy of the fit and the computational time was made. Table 1 lists the values of <k> and N_g_/N obtained from our experiments and simulations. The C_1_ and C_2_ values resulting in <k> and N_g_/N values are also reported in Table 1, indicating a clear decrease in the ratio between tip-to-tip fusion and fission rates (a_1_ and b_1_ in C_1_ = a_1_/b_1_). In fact, these results indicated that the ratio between tip-to-tip fusion and fission rates was about 83% smaller in ATRA-treated cells compared to the control cells. We also report the fraction of X_1_, X_2_, and X_3_ species in the network at steady state in Table 2 for control and ATRA-affected cells. We found a smaller number of X3 species in ATRA-treated cells, which further confirmed the greater mitochondrial fragmentation in ATRA cells compared to control cells. In Figure 4E, the number of mitochondria is shown to have been increased by ATRA treatment (Panel E, * *p* < 0.05 vs. control). Such an increase could be attributed to the increase in fragmentation or fission (lower X3), as suggested by the simulations. Similarly, in Figure 5, it is seen that ATRA decreased the mitochondrial loop size (i.e., decreased number of mitochondria in a loop), branch length (number of edges in a linear branch), and mitochondrial cluster size (number of edges in a cluster, which consists of both linear branches and cyclic loops but is disconnected from the rest of the network). The simulations of Figure 6 assess the structure of the mitochondrial network (regarding whether the network was fragmented) regardless of the network size. That is, Figure 6 gives a measure of how fragmented the mitochondrial network that was imaged and quantified in Figure 5 was and what the microscopic fusion and fission rates that could explain such fragmentation were. The simulations indicated that the ratio between tip-to-tip fusion and fission rates was about 83% smaller in ATRA-treated cells compared to control cells.

## 4. Discussion

Mitochondrial fission is a dynamic event that is regulated by proteins such as DRP1 and FIS1. Normal fission is necessary for maintaining cellular function. The fission machinery is vital for tissue development. For instance, DRP1 knockout in mice is embryonically lethal [73], and aging leads to decreased DRP1 protein levels in cardiac and skeletal muscles [74]. Previously, Vantaggiato C et al. tested the effects of ATRA in embryonic carcinoma cell lines (P19 cells) [43]. P19 cells were differentiated into neurons and glial cells in the presence of 5 μM ATRA for 14 days. It was found that DRP1 was upregulated during the neuronal differentiation by ATRA [43]. However, it is not known whether ATRA can also upregulate cardiac DRP1. Here, we tested in vivo and in vitro whether ATRA affects DRP1 in the heart. Our results suggested that ATRA elevates DRP1 levels and elicits mitochondrial network remodeling by increasing mitochondrial fission, as suggested by our numerical modeling. Our results showed that ATRA dose-dependently increased DRP1 levels, decreased mitochondrial area and perimeter, and increased circularity.

Mitochondria are central players in cell metabolism and bioenergetics, and their dysfunction could underlie disease processes [75]. Robust mitochondrial fission and fusion events sustain an efficient and healthy mitochondrial matrix [73]. Therefore, the proteins and factors involved in mitochondrial fission–fusion are being explored as therapeutical targets [74,76]. The modulations of these events through post-translational modifications, cofactors, and externally applied agents could result in improving mitochondrial biogenesis and the organelle’s network integrity.

DRP1 is an integral mitochondrial fission factor. Transcriptional [77,78] and post-translational modifications [79] are key mechanisms regulating and controlling DRP1 levels. The loss or dysfunction of DRP1 has been shown to be lethal during embryonic development [80]. DRP1 downregulation has been implicated in apoptosis [81] and mitochondrial autophagy [82], causing increased oxidative stress levels and leading to a cascade of inefficient and detrimental cellular processes.

It has been proposed that promoting a balanced and healthy mitochondrial network by targeting DRP1 could be a useful approach that improves cellular energetics and metabolism in different pathologies [83]. Studies used the genetic overexpression of DRP1 as a way to increase mitochondrial fission [83,84,85,86]. Here, we evaluated the effect of pharmacologically mediated DRP1 increase via ATRA. Our experiments in HEK293 and HL-1 cell lines, as well as in mice, suggested that ATRA results in increased DRP1 levels and mitochondrial remodeling.

Mitochondrial network integrity is balanced by fission and fusion processes [87], and a healthy mitochondrial network is essential for cellular function [88]. Our mitochondrial network analysis showed that, compared to vehicle control-treated cells, ATRA-treated HL-1 cells possess a lower mean degree (<k>), N_g_/N, smaller loops/cluster sizes, and smaller branch lengths representative of fragmented networks. Our stochastic simulations further suggested that at the microscopic level, mitochondrial fusion and fission dynamics are shifted by ATRA. Specifically, an upregulated tip-to-tip fission reaction could explain the mitochondrial network analysis results.

Various findings have shown that DRP1-mediated processes are crucial for maintaining normal cardiac function [89,90]. It was shown that DRP1 inhibition decreased cell death in HL-1 after simulated ischemia/reperfusion [91]. Ashrafian et al. found that alteration in the DRP1 gene leads to the enlargement of ventricular thickness, contractility, and increased fibrosis in transgenic mice [92]. Tang et al. found a lower level of DRP1 in chronically hypertrophied, hypertensive rat hearts [93], but the pro-hypertrophic neurohormone norepinephrine was found to increase DRP1 levels in cardiomyocytes [94]. Additionally, it was shown that DRP1 downregulation resulted in mitochondrial elongation, impaired mitochondria aggregation, and accelerated apoptosis in cardiomyocytes [89]. Several studies have shown that the loss of DRP1 enlarges the heart size by possibly decreasing autophagy and mitophagy [95,96,97].

To our knowledge, there is no known pathway directly linking ATRA to increased DRP1 levels. It could be possible that ATRA induces epigenetic modifications through retinoic acid receptor-dependent mechanisms that ultimately modulate DRP1 expression [98].

Our results indicated that while ATRA treatment increased DRP1 levels, it did not alter the levels of the mitochondrial fusion protein OPA1. OPA1 is essential for maintaining normal fusion, as hampering mitochondrial fusion can harm the cristae structure and inner mitochondrial membrane integrity, leading to the release of cytochrome c [99,100].

## 5. Limitations

The accuracy of the estimated fusion rates, fission rates, and mitochondrial network statistics using the image processing and modeling approaches used in this study relied on the confocal imaging of the mitochondrial network. The construction of the digitized mitochondrial network from experimental images relied on image segmentation steps such as image smoothing, threshold selection, background noise removal, thinning, and skeletonizing. All these parameters could potentially affect size, structure, and microscopic parameters such as mean network degree <k>, giant cluster size Ng, Ng/N ratio, the distribution of cluster sizes, branch lengths, and cell cycles of the identified network. The final network generated by the model was also sensitive to the intrinsic stochastic nature of the fission and fusion processes and could vary depending on the duration for which a simulation was run and the number of times a simulation with a given parameter set was repeated. We overcame these limitations by performing long simulations to ensure that an actual steady state was reached and performing hundreds of repetitions for each control and ATRA-treated cell. The total mitochondrial volume in ATRA versus vehicle control treatment was not studied. Our imaging of the mitochondrial network in HEK293 cells did not result in robust images that could be used for analysis, as done in Figure 5. Therefore, only HL-1 cells were used.

## 6. Conclusions

Our findings suggested that ATRA elevates DRP1 levels without changing those of OPA1, promotes fission events, alters the mitochondrial network, and modifies the ultrastructure of mitochondria in the heart. Future investigations should assess whether ATRA could play a role in cardiac protection by eliminating the damaged parts of mitochondria through a DRP1-mediated mitochondrial fission process.

## Figures and Tables

**Figure 1 cells-10-01202-f001:**
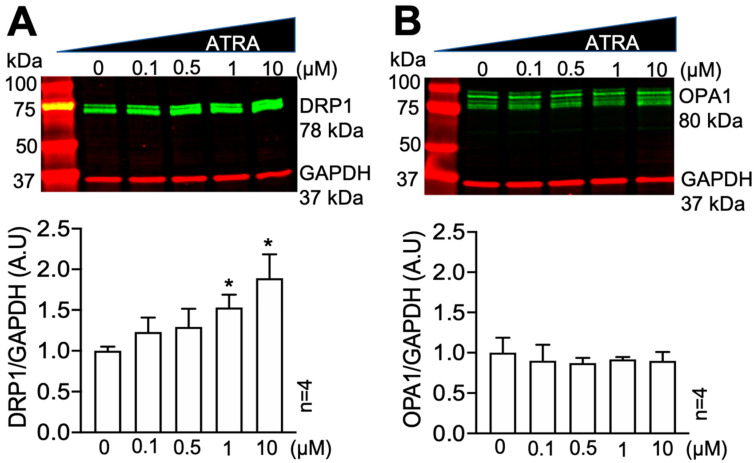
Dose-dependent effect of ATRA on DRP1 and OPA1. Western blot of DRP1 protein levels in HEK293 cells incubated with 0.1, 0.5, 1, and 10 μM ATRA for 24 h, normalized to the levels of DMSO controls (0 μM). (**A**) ATRA increased the levels of DRP1 in a dose-dependent manner (* *p* < 0.05, 1, and 10 μM ATRA vs. DMSO control (0 μM); One sample *t*- and Wilcoxon-test). (**B**) ATRA did not affect OPA1 levels.

**Figure 2 cells-10-01202-f002:**
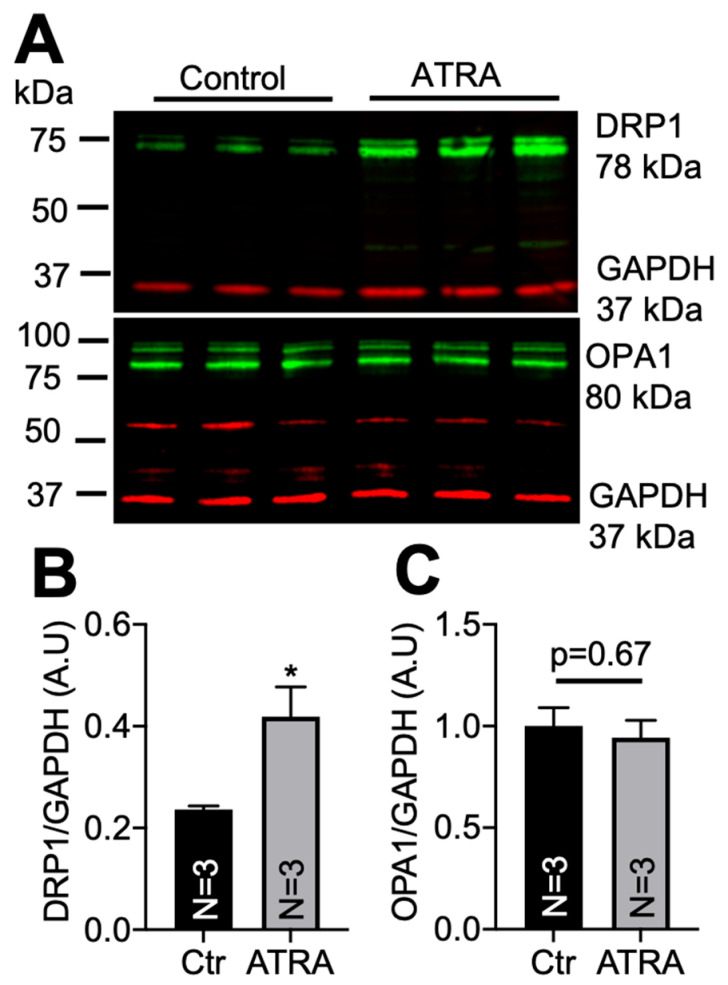
DRP1 and OPA1 levels in mouse hearts treated with an i.p. injection of ATRA or corn oil (vehicle control). (**A**) Western blots of DRP1 (upper panel) and OPA1 (lower panel). (**B**) DRP1 levels were significantly higher in ATRA-treated mice (* *p* < 0.05, vs. vehicle; unpaired 2-sample *t*-test). (**C**) There was no effect of ATRA on OPA1.

**Figure 3 cells-10-01202-f003:**
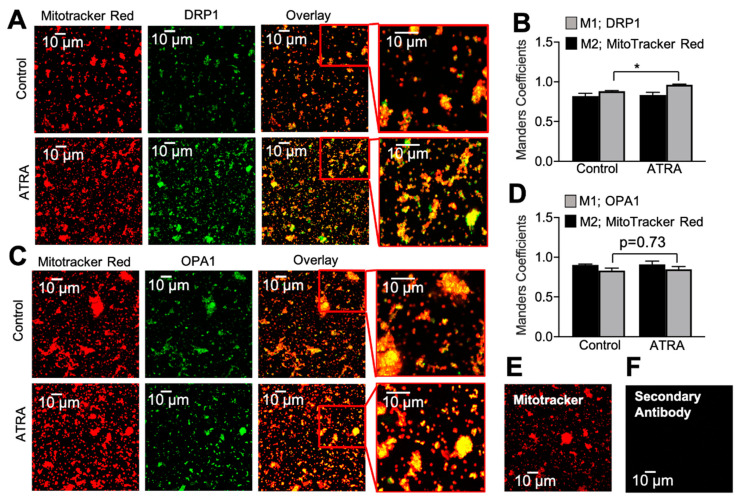
Quantification of mitochondrial localization of DRP1 and OPA1 by confocal microscopy. (**A**) Mitochondria were labeled with antibodies against DRP1 and (**C**) OPA1 in purified cardiac mitochondrial clusters from vehicle- (upper panels) and ATRA- (lower panels) treated mice. Red channel: MitoTracker™ Red. (**B**) and (**D**) Mander’s coefficient of colocalization between MitoTracker™ Red (M2; black bars) and DRP1 (M1; gray bars) vs. OPA1 (M1; gray bars), respectively. In ATRA treatment, there was a significant increase in DRP1 colocalization with MitoTracker™ Red (* *p* < 0.01, vs. vehicle; unpaired 2-sample *t*-test; *n* = 3 mice; *n* = 12 images) (**B**) but no change in OPA1 (*n* = 3 mice; *n* = 12 images) (**D**). Mitochondria were loaded with MitoTracker™ Red (**E**) and labeled with secondary antibody alone (**F**).

**Figure 4 cells-10-01202-f004:**
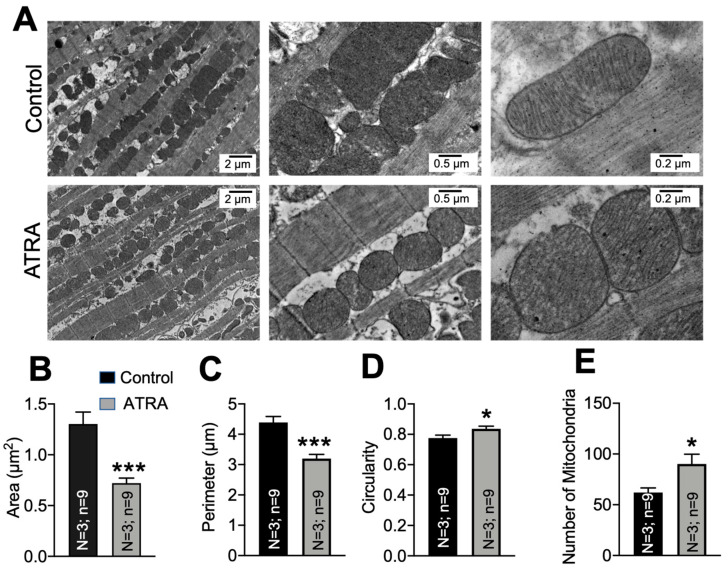
Effects of ATRA on mitochondrial ultrastructure in mice treated with ATRA versus control. (**A**) Representative electron micrographs from ATRA- and corn oil- (vehicle) treated mice at 10,000×, 25,000×, and 50,000× magnification. Mitochondrial (**B**), average area (**C**), perimeter and (**D**) circularity, and (**E**) number of mitochondria per area. Mitochondrial area and perimeter were significantly decreased (**B**,**C**) and mitochondria were more circular in the ATRA-treated group (**D**) (* *p* < 0.05, *** *p* < 0.001, vs. vehicle; unpaired 2-sample *t*-test). *n* = 3 hearts; *n* = 9 micrographs each condition.

**Figure 5 cells-10-01202-f005:**
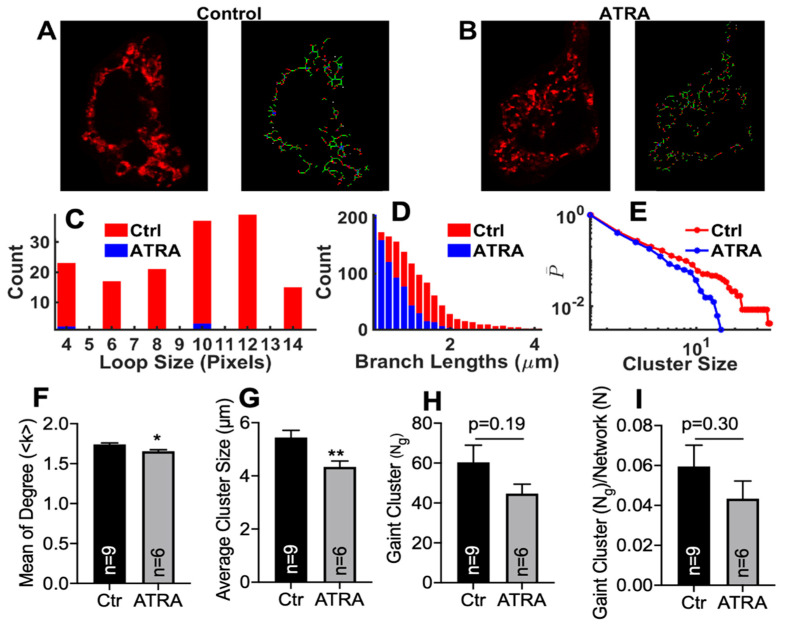
Microscopic properties of the mitochondrial network in control and ATRA-treated HL-1 cells. Representative confocal images of the mitochondrial network in (**A**) control and (**B**) ATRA-treated cells from experiments (left panels) and the network retrieved using image processing (right panels). Distributions of loop sizes (**C**), branch lengths (**D**), and cluster sizes (**E**) (cumulative probability) for control (red) and ATRA-treated cells (blue). Mean degree (**F**), average cluster size (**G**), giant cluster (**H**), and the giant cluster normalized to the total network size (**I**). * *p* < 0.05, ** *p* < 0.01, vs. control; unpaired 2-sample *t*-test.

**Figure 6 cells-10-01202-f006:**
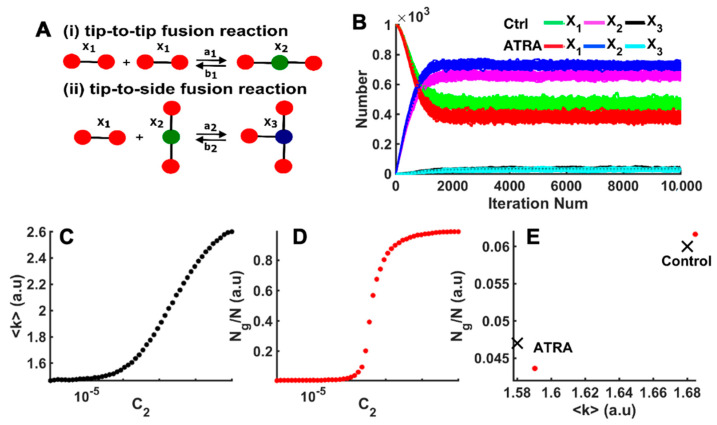
Estimating relative fusion and fission rates using agent-based model. (**A**) Model scheme representing the tip-to-tip fusion of two X_1_ nodes into X_2_. Tip-to-side fusion of one X_1_ node with one X_2_ node to form one X_3_ node, as well as the corresponding fission processes. (**B**) The number of vehicle (control), or ATRA X_1_, X_2_, and X_3_ species from the model as functions of the number of iterations using the C_1_ and C_2_ values given in Table 1. Model results for mean degree (**C**) and N_g_/N (**D**) as functions of C_2_ at fixed C_1_ = 0.0007 and *n* = 3000. (**E**) Comparison of N_g_/N versus <k> values obtained from experiment (red bullets) and simulation (black crosses) in control and ATRA-treated cells.

**Table 1 cells-10-01202-t001:** Comparison of microscopic parameters of mitochondrial network obtained from experiments (Exp) and simulations (Theory). Mean degree (columns 2 and 3) and Ng/N (columns 4 and 5) from experiment and theory. Columns 6 and 7 list the values of C1 (tip-to-tip fusion/fission ratio) and C2 (tip-to-side fusion/fission ratio) obtained by fitting the model to the data. Notice that C1 decreased by 83% in ATRA-treated cells compared to control.

Condition	Mean Degree <k>		N_g_/N		C_1_	C_2_
	Exp	Theory	Exp	Theory		
Ctrl	1.68	1.68	0.060	0.062	5.5 × 10^−3^	2.0 × 10^−4^
ATRA	1.58	1.59	0.047	0.045	3.0 × 10^−3^	2.0 × 10^−4^

**Table 2 cells-10-01202-t002:** The fractions of X1, X2, and X3 species at steady state using C1 and C2 values estimated for control and ATRA-treated cells. Notice that X1, X2, and X3 decreased by 25%, increased by 17%, and decreased by 10%, respectively, in ATRA-treated cells compared to control.

Condition	X_1_	X_2_	X_3_
Ctrl	0.436	0.541	0.023
ATRA	0.349	0.631	0.020

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
