# Peer review of "All-Trans Retinoic Acid Increases DRP1 Levels and Promotes Mitochondrial Fission"

_cells, 2021, doi:10.3390/cells10051202_

Round 1

Reviewer 1 Report

The authors present a combined experimental and modeling study of the effect of retinoic acid on DRP1, OPA1 and mitochondrial morphology.  There are several issues that the author need to address.

  1. It is not clear that the authors fully are cognizant of the previous work in this area.  In https://www.ncbi.nlm.nih.gov/pmc/articles/PMC6458285/ Incubating with retinoic acid increased DRP1 levels in neurons.  This should be discussed as it is one of the main findings of the paper.  A clear delineation of what is new should be presented.
  2. In Figure 1, how was the statistical significance calculated, what was being compared.
  3. In the experiments of Figure 4, was the total mitochondrial volume conserved?
  4. The explanation of the model is almost impossible to understand. I read it a few time and still it is not clear. The authors should use the standards in the field of presentation model formulation.
  5. Line 378 the meaning of <k>, Ng, Ng/N are not defined before the appear. I believe they are defined later in the paragraph.
  6. It is not clear how the model related to experimental results. In Figure 4  and 5 ATRA seems to increase the number of mitochondria and decrease the size.  However in figure 6 it seems the opposite.  This needs to be clarified.
  7. It is not clear what insights the model yields.  Does the model actually add anything to the paper?
  8. The discussion is mostly a summary. The last paragraph does refer to the literature but it is really not a discussion.

Author Response

We thank the editor for the opportunity to submit a revised version of our manuscript “All-Trans Retinoic Acid increases DRP1 levels and promotes mitochondrial fission”. We appreciate the time and effort that the editor and the reviewers dedicated to providing suggestions for improvement. We have revised the manuscript, and as a result of the reviewers’ suggestions which we addressed, the work has been greatly improved. We highlight in yellow the changes that were made to the manuscript. Please see below for a point-by-point response to the reviewers’ comments and concerns.

Reviewer 2 Report

In the manuscript entitled, “All-Trans Retinoic Acid increases DRP1 levels and promotes mitochondrial fission”, Chidipi, et al., have investigated the effect of ATRA on mitochondrial dynamics in vitro and in vivo using a variety of tools including confocal microscopy, electron microscopy, and stochastic modeling. The authors conclude that treatment with ATRA results in increased expression and mitochondrial localization of the mitochondrial fission protein, Drp1. The increased Drp1 is further shown to alter mitochondrial ultrastructure which has been validated by confocal microscopy in combination with stochastic modeling. The role of ATRA in regulating one of the most important facets of cardiac cell functioning via mitochondrial remodeling is interesting. In addition, the development of stochastic mitochondrial network remodeling studies would provide an opportunity for studying mitochondrial physiology in other diseases including cancer, aging, and metabolic disorders. The authors should address the following concerns before the manuscript can be accepted for publication.

  1. The data for the effect of ATRA on MFN1 and MFN2 should be included.
  2. In Figure 2A, although the induction of Drp1 upon Atra treatment is evident, it seems the levels of GAPDH are higher in ATRA-treated samples compared to control samples. The authors must cross-check the densitometry-based quantification data and replace the blot with a more representative one.
  3. In figure 2, the authors must include the effect of ATRA on other mitochondrial fusion/fission proteins.
  4. In the legend for figure 3, the authors must correct the typographical error in the spelling for ATRA.
  5. Figure 3B and 3D, the details of the statistical test applied must be included in the legend. The authors must also include the number of cells that were analyzed to compute the Mander’s coefficient.
  6. Figures 3G, H do not make much sense and must be removed. It would be logical to include the TEM images of mitochondria treated with vehicle or ATRA (which the authors have included in figure 4).
  7. The modeling of the mitochondrial network was performed in HL-1 cells instead of the previously used cell line model, HEK293. Is there a particular reason why the authors have switched to the cardiac cell line? The authors must, in fact, include the data for HEK293T and HL-1 cell lines in both figure 1 and figure 5 to emphasize the robustness of the model.
  8. In figure 5, the details of the statistical test applied must be included in the legend.
  9. While the authors have developed a very interesting simulation method to compute the mitochondrial fission and fusion rates, they must include if the values for different parameters derived using the simulations are significant in Tables 1 and 2.
  10. The effect of ATRA on mitochondrial dynamics is a very interesting facet to regulate cellular metabolism and physiology. Can the authors comment on a plausible mechanism for ATRA-induced Drp1 expression resulting in mitochondrial fission?
  11. Can the authors also comment on the effect of heightened mitochondrial fission on the survival of cells upon ATRA treatment as cellular metabolism is also dependent on mitochondria? Also, did the authors observe an increase in autophagy, specifically, mitophagy, upon ATRA treatment?

Author Response

We grateful for the reviewer’s encouraging comments and for the valuable suggestions. Performing the requested corrections and addressing the raised concerns have helped us present a better and stronger work.

Round 2

Reviewer 1 Report

The authors have addressed my concerns.

Minor points

line 169 should not be indented.

line 550 is a one sentence paragraph.  The sentence should probably not be alone in a paragraph

Reviewer 2 Report

The authors have addressed all the raised concerns satisfactorily. The manuscript should be accepted for publication.